# MULTISCALE POSITIVE-UNLABELED DETECTION OF AI-GENERATED TEXTS

**Yuchuan Tian**[1], **Hanting Chen**[2], **Xutao Wang**[2], **Zheyuan Bai**[2], **Qinghua Zhang**[3],
**Ruifeng Li**[4], **Chao Xu**[1], **Yunhe Wang**[2]*

[1] National Key Lab of General AI, School of Intelligence Science and Technology, Peking University
[2] Huawei Noah's Ark Lab [3] Huawei Group Finance [4] Huawei Central Software Institute
tianyc@stu.pku.edu.cn, yunhe.wang@huawei.com

## ABSTRACT

Recent releases of Large Language Models (LLMs), e.g. ChatGPT, are astonishing at generating human-like texts, but they may impact the authenticity of texts. Previous works proposed methods to detect these AI-generated texts, including simple ML classifiers, pretrained-model-based zero-shot methods, and finetuned language classification models. However, mainstream detectors always fail on short texts, like SMSes, Tweets, and reviews. In this paper, a Multiscale Positive-Unlabeled (MPU) training framework is proposed to address the difficulty of short-text detection without sacrificing long-texts. Firstly, we acknowledge the human-resemblance property of short machine texts, and rephrase AI text detection as a partial Positive-Unlabeled (PU) problem by regarding these short machine texts as partially "unlabeled". Then in this PU context, we propose the length-sensitive Multiscale PU Loss, where a recurrent model in abstraction is used to estimate positive priors of scale-variant corpora. Additionally, we introduce a Text Multiscaling module to enrich training corpora. Experiments show that our MPU method augments detection performance on long AI-generated texts, and significantly improves short-text detection of language model detectors. Language Models trained with MPU could outcompete existing detectors on various short-text and long-text detection benchmarks. The codes are available at https://github.com/mindspore-lab/mindone/tree/master/examples/detect_chatgpt and https://github.com/YuchuanTian/AIGC_text_detector.

## 1 INTRODUCTION

Recent developments in Large Language Models (LLMs) have brought astonishing changes to people's lives. The GPT-2 (Radford et al., 2019) model, created in early 2019, is capable of simple question-answering tasks; GPT-3 (Brown et al., 2020) is a great leap in model size and capability; ChatGPT (OpenAI, 2022), announced in late 2022, shows comparable performance to humans as a chatbot; GPT-4 (OpenAI, 2023a), released this year, has even better generative performance. These advancements are making people's lives easier with applications like writing aids, search engines, and Office Suites. However, they could be used to generate deceptive fake texts for illegal and unethical purposes.

Previous works have proposed numerous approaches to distinguish fake AI-generated text from genuine human languages. Canonical work (Solaiman et al., 2019) used simple machine learning classifiers as baselines; some works (Gehrmann et al., 2019; Mitchell et al., 2023) proposed zero-shot detection measures based on pretrained models; numerous works (Solaiman et al., 2019; Crothers et al., 2022; Guo et al., 2023; Mitrovic et al., 2023) perform simple finetuning of pretrained language models on the AI-text classification task.

Despite various methods, few mainstream methods investigated the negative impact of text length: the difficulty to detect significantly increases as texts become shorter. Some latest online ChatGPT detectors have noticed this issue, but they dodge rather than address it by putting up minimum text

---
*Corresponding Author.

length requirements (Tian, 2022; FudanNLPLab, 2023; OpenAI, 2023b). In the era of smartphones where people rely heavily on fragmented mobile media, fake short articles like SMSes, Tweets, and reviews generated by LLMs could pose huge threats to one's daily life, yet we still lack a comprehensive detector that is capable of detecting both short texts and long-texts.

To improve detectors' performance on short texts, we rethink the plain "Binary Classification" setting that is intuitively applied. It is seemingly natural to phrase text detection as a binary classification task, as texts have clear origins (from human works or AI outputs) and thus, clear binary labels (real or fake); but interestingly, we observe a handful of machine-generated texts that are overly short and simple, such that these texts are highly similar to human (*e.g.* Ex. 2 in Table 1). It is not suitable to assign these simple machine texts with either clear human or AI labels; rather, they are in an "Unlabeled" state. Though the case is occasional and most short machine texts (*e.g.* Ex. 1 in Table 1) are still distinguishable based on manifold features, it prompts us to question the rationality of clear binary labels on general short machine texts. On the contrary, we hold that short machine-generated texts are partially "Unlabeled". As machine-generated texts become shorter and simpler, the "Unlabeled" property could gradually dominate the text.

| **Example 1**: The first sentence in benchmark HC3-Sent (Guo et al., 2023) | |
|---|---|
| **Human:** You can't just go around assassinating the leaders of countries you don't like! | **AI:** It is generally not acceptable or ethical to advocate for or condone the assassination of any individual, regardless of their actions or beliefs. |
| **Example 2**: Answer to "When is the independence day of the United States?" | |
| **Human:** Independence Day is annually celebrated on July 4th. | **AI:** The Independence Day of the United States is celebrated on July 4th. |

Table 1: Short example answers from human and AI. In general, short answers are distinguishable based on features like punctuations, emotions, and formality (see non-cherrypicked case Ex. 1). But in extreme cases (see Ex. 2), short simple answers are indistinguishable, and the unlabeled property is manifest.

In this sense, we model the task of AI-generated text detection as a partial Positive-Unlabeled (PU) problem and formulate the Multiscale Positive-Unlabeled (MPU) training framework to address the challenging task of short text detection without sacrificing long texts. PU problems typically address binary classification tasks where positive data and unlabeled data are offered for training. Considering the partially "Unlabeled" property of short machine texts, we rephrase detector training as a partial PU problem and boost detectors' performance on multiscale texts. In order to improve conventional PU optimization targets for texts of various lengths, a length-aware Multiscale PU (MPU) loss is proposed and applied during the training process. We are aware that the PU prior probability of a text being positive is length-variant. To this end, an abstract recurrent model is designed to adjust the PU prior probability automatically based on corpus length. Further, a Text Multiscaling module is also proposed to exert the effect of Multiscale PU loss by diversifying training corpora in terms of length. Experiments demonstrate that the MPU framework is significantly effective in improving short-text detection performance; meanwhile, detection on long texts is also augmented.

## 2 RELATED WORK

**Text Detection Methods.** Since the introduction of GPT-2 (Radford et al., 2019) and its successors, fake texts generated by powerful LLMs are causing ethical and legal issues. Methods are developed to discriminate against these generated texts in various misuse scenarios. Zellers et al. (2019) shed light on machine-generated fake news by proposing a GPT-based news generator GROVER, and uses GROVER itself to sort fake news out; Adelani et al. (2020) looks at detection of fake online reviews; Fagni et al. (2020) focuses on machine-generated fake tweets and proposes the TweepFake dataset. Other proposed detection methods are for general scenarios. Several canonical baselines are mentioned by Solaiman et al. (2019) to detect GPT-2 texts, including simple TF-IDF classifiers and finetuned RoBERTa (Liu et al., 2019); GLTR (Gehrmann et al., 2019) detect generated texts in a zero-shot manner by using token prediction probabilities from available pretrained NLP models like BERT (Devlin et al., 2018) and GPT-2 (Radford et al., 2019). After the introduction

of ChatGPT (OpenAI, 2022), some new detection methods (Liu et al., 2022; Mitchell et al., 2023; Mitrovic et al., 2023; Guo et al., 2023) are released.

**PU Methods.** Previous works have proposed methods to train a binary classifier with positive and unlabeled data. Many PU methods (Bekker & Davis, 2020; Du Plessis et al., 2014; Kiryo et al., 2017; Su et al., 2021; Hammoudeh & Lowd, 2020; Chen et al., 2020) constructs PU loss based on positive and unlabeled samples, for classifying unlabeled data. Other PU methods include two-step learning and bias learning (Liu et al., 2003). The two-step technique first identifies reliable negative examples and then performs learning based on the positives and negatives of the mark (He et al., 2018; Ienco & Pensa, 2016); biased learning treats unlabeled data as a negative sample of class-labeled noise (Hsieh et al., 2015; Shao et al., 2015). Above all, we refer to applying a PU loss during training to address the task of multiscale AI-generated text detection, because PU losses could be generally applied on powerful finetuning text detectors without much additional computation costs.

# 3 MULTISCALE POSITIVE-UNLABELED TEXT DETECTION

## 3.1 TEXT DETECTION AS POSITIVE-UNLABELED CLASSIFICATION

Despite manifold methods for detecting AI-generated texts, mainstream detectors seldom take the factor of text length into account, and thus they always fail on short texts. We have tried several existing detection methods for short LLM-generated texts (shown in Table 4), but none of them perform well. As people nowadays are immersed in short, fragmented forms of mobile media, they are vulnerable to LLM attacks with no reliable means to defend themselves. Hence, we are in urgent need of a performant short AI-generated text detector.

Intuitively, past works formulated the task of AI text detection as a binary classification problem, *i.e.* classifying texts as AI or Human. However, the formulation could be problematic for shorter texts as we found high similarities between extremely simple AI texts and human texts. The phenomenon could be rare in actual applications. But it is fundamentally reasonable, because LLMs learn from human languages; and for sentences whose structures are overly simple, they are seemingly "copied" by LLMs from what they have learned. Therefore, the attribution of these simple machine texts is uncertain: on one hand, they are indeed outputs from Language Models; on the other hand, they are ordinary human languages. Though the completely non-classifiable case mostly happens for extremely short texts or commonly used phrases (that rarely occurs in our benchmarks and detection of which is of no application value), it inspires us to think about the partially "unlabeled" property behind the vast majority of short, distinguishable texts despite their definite labels.

To overcome this issue, we model the task of multiscale text detection as a partial Positive Unlabeled problem (PU). In this problem, corpora from human are regarded as "Positive", but short texts from machines are given an additional "Unlabeled" mark for PU loss calculations (detailed in Sec. 3.3). Then our detector model is optimized within this partial PU context.

## 3.2 PRELIMINARIES: CANONICAL PU LOSS FUNCTIONS

PU losses are derived from the traditional Positive-Negative (PN, *i.e.* Binary Classification) setting, detailed in Appendix A. Some works (Du Plessis et al., 2014; Plessis et al., 2015) perform indirect approximation of the negative risk in the PN framework, yielding the unbiased PU (uPU) loss as follows:

$$\hat{R}_{uPU}(g) = \pi \hat{R}_P(g, +1) - \pi \hat{R}_P(g, -1) + \hat{R}_U(g, -1), \qquad (1)$$

where $\hat{R}_P(g, -1) := \frac{1}{n_P} \sum_{i=1}^{n_P} L(g(x_i^P), -1)$ and $\hat{R}_U(g, -1) := \frac{1}{n_U} \sum_{i=1}^{n_U} L(g(x_i^U), -1)$ are estimations calculated from positive and unlabeled training samples respectively.

However, the deep learning classifier may be too flexible, leading to $\hat{R}_U(g, -1) - \tilde{\pi} \hat{R}_P(g, -1) < 0$ and causing the model to overfit. As a remedy, Kiryo et al. (2017) proposes the non-negative risk estimator based on the uPU loss. The non-negative PU (nnPU) loss is thus derived as follows:

$$\hat{R}_{nnPU}(g) = \tilde{\pi} \hat{R}_P(g, +1) + \max\{0, \hat{R}_U(g, -1) - \tilde{\pi} \hat{R}_P(g, -1)\}. \qquad (2)$$

The nnPU loss Kiryo et al. (2017) is performant and thus widely referred by later PU works and applications (Kato et al., 2019; Bepler et al., 2019; Peng et al., 2019; Xu et al., 2019; Chen et al., 2020; Su et al., 2021; Tang et al., 2022). However, to the best of our knowledge, no previous works have applied PU to scenario of length-variant texts, in which simple usage of the nnPU loss might not be effective. We hope to develop an effective PU mechanism in aid of detecting length-variant texts.

### 3.3 MPU: A Length-sensitive PU Approach

In PU loss conventions as stated in Sec. 3.2, the estimation for the prior probability of a data being positive $\tilde{\pi}$ is always kept at a constant. The reason is that prior probability $\pi$ is closely associated with the dataset distribution, which is always assumed to be uniform. However, this might not be case with texts of different lengths. As explained in Section 1, short texts and long texts hold different properties; in other words, they do not share the same distribution. In this regard, the assumption of dataset distribution being uniform is flawed; fixing the prior estimation at a certain constant value is problematic in the case of multiscale text detection (*i.e.* where texts to be processed are of manifold length).

Though long texts and short texts have different distributions, the distribution shift from long text to short text is a gradual process with respect to text lengths. To deal with the gradual shift of distribution, we look at this shift with respect to text length from a differentiation perspective. Texts of a certain length $l$ could be regarded as a small subset that features its own distribution, and also its own prior $\pi(l)$. We hope to provide a smooth, length-variant estimation $\tilde{\pi}(l)$ for the prior at length $l$, in order to fit the PU framework for the multiscale text detection problem.

In this fashion, we propose the Multiscale PU loss $\hat{R}_{MPU}$ that uses length-sensitive priors $\tilde{\pi}$ for multiscale texts. However, we are faced with the challenge of modeling the length-variant prior $\tilde{\pi}$ in abstraction. Namely, we need to investigate the general probability of all sentences (of a certain length) being human, without access to specific details of any piece of text. To this end, we use the general recurrent language model (Mikolov et al., 2010; Sundermeyer et al., 2012) in abstraction as a discriminator for positive, human-spoken corpora, which is formulated as follows: given a sequence $S_l$ of $l$ tokens: $S_l = [t_i]_{i=1}^n$, abstract recurrent discriminator $\Delta : seq \to [0, 1]$ that is bounded one-dimensional (because from the discriminator we expect a confidence of a sequence being positive), the recurrent model in abstraction is expressed as:

$$\Delta(S_{i+1}) = f(\Delta(S_i), t_{i+1}), \forall i \in [l-1], \tag{3}$$

where $f$ is some function that merges the classification of all previous tokens $S_{i-1}$ with the classification of the last token $t_i$. Next, the abstraction is concretized based on task characteristics of human-generated text discrimination. Since relatively short texts tend to have simple semantic correlations to be captured, human text discrimination is performed via capturing signals from tokens. We hold that each token has a hidden property of origin, and the attribution contributes to the classification of the whole sequence. Tokens, as extreme cases of short texts, could be sorted into two categories: "clear positive", *i.e.* the token could hardly be generated by AI; or "unlabeled", i.e. the token is mediocre and universally used, giving no signal as "human-spoken". Each token is expected to provide an equal contribution to the overall sequence classification towards the orientation of its own category (Kang et al., 2018). In this sense, the merging function $f$ is formulated as equally-weighted addition:

$$f(\Delta(S_i), t_{i+1}) = w_S \Delta(S_i) + w_t \delta(t_{i+1}) \quad \text{s.t.} \quad w_S = w_t, \tag{4}$$

where $\delta(t_{i+1})$ is defined as the contribution of $\delta(t_{i+1})$. For simplicity, we discretize the transition of classification from $i \to i+1$ and each token contribution is designated as binary. We also take text length into consideration by normalizing $\delta(t_{i+1})$ with a factor of sequence length $l$. Under these assumptions, the transition is formulated as:

$$\Delta(s_{i+1}) = \text{clip}(\Delta(S_n) + \delta(t_i), [0, 1]), \quad \text{s.t.} \quad \delta(t_i) = \begin{cases} 1/l & \text{if } t_i \text{ is clear positive,} \\ -1/l & \text{otherwise.} \end{cases} \tag{5}$$

Notably, we use a hard clip function to bound the overall classification results in interval $[0, 1]$ rather than other non-linear functions, e.g. sigmoid. This is because clear positive tokens could be rare in

practice. This assumption is particularly true when we consider recent advancements of generative language models, where human and AI languages are more resembling. In other words, a majority of words are both frequently used by human and AI, while only a few signal words manifest unique human characteristics. This property requires the discriminate model to be highly sensitive to positive token signals. Hence, we set hard boundaries rather than using non-linear standardizing functions to scale the output between $[0, 1]$. Further, to encourage positive responses, we initially positive as the initial state $\Delta(S_0)$ of the discriminator.

Return to the original objective, we tend to calculate the prior probability of a sample being positive $\tilde{\pi}$ based on the introduced recurrent language model. $\tilde{\pi}$ could also be interpreted as the expectation of confidence from the recurrent discriminator $E\left[\Delta(S_l)\right]$. The discretization of contribution is beneficial to reducing the continuous discriminator $\Delta$ to discrete states: for a sequence $S_l$ with $l$ tokens, the confidence could only take values as $i/l, \forall i \in [l]$. Therefore, discriminator $\Delta$ has a total of $i + 1$ equally spaced states as confidence output. We will show that the expectation $E\left[\Delta(S_l)\right]$ of all length-$l$ sequences could be exactly calculated given the positive probability $p$ of a single token, i.e. the general probability of a token showing clear-human signal. As stated previously, $p$ tends to be a small value. State transition matrix $\mathbf{P} \in \mathbb{R}^{(l+1)\times(l+1)}$ that represents the contribution of the last token is a band sparse matrix consisting of positive transition $p$ and negative transition $1 - p$ to adjacent states from the current state. Defining probability vector at state $i$ as $\sigma_i \in \mathbb{R}^{(l+1)}$, a single transition shown as Eq.5 and the final state probability vector could be described as:

$$\sigma_{i+1} = \sigma_i \mathbf{P}, \quad \sigma_l = \sigma_0 \mathbf{P}^l. \tag{6}$$

Thus, given one-hot initial state $\sigma_0$, we could calculate the final state probability vector and the overall expecation $\tilde{\pi}$ for a sequence of length $l$:

$$\tilde{\pi}(l) = E\left[\Delta(S_l)\right] = \langle \sigma_l, \alpha \rangle = \sigma_0 \mathbf{P}^l \alpha^T, \tag{7}$$

where vector $\alpha \in \mathbb{R}^{(l+1)}$ is the sequence vector of all possible positive confidence: $\alpha = [i/l]_{i=0}^l$. Further details and derivations are mentioned in Appendix B. As a result, as text length decreases, the prior positive probability in samples of this length $\tilde{\pi}_{length}$ decreases as well. This is in line with our expectation in Sec 3.1 that shorter texts tend to demonstrate more "unlabeled" properties.

Finally, on top of the canonical non-negative PU loss as defined in Eq. 2, we define the Multiscale PU Loss with text-length-variant priors:

$$\hat{R}_{MPU}(g) = \langle \tilde{\Pi}, \hat{R}_P(g, +1) \rangle + \hat{R}_U(g, -1) - \langle \tilde{\Pi}, \hat{R}_P(g, -1) \rangle, \tag{8}$$

where $\tilde{\Pi}$ stands for an array: $[\tilde{\pi}(l_g)]$ that records the corresponding prior of training texts, calculated based on respective text lengths using Eq. 7. As is emphasized, short machine-generated texts should be viewed as partially "unlabeled" rather than entirely "unlabeled". Hence, we weight-sum the multiscale PU loss and the canonical PN classification loss to get the final loss for detector model finetuning:

$$\hat{R}(g) = \hat{R}_{PN}(g) + \gamma \hat{R}_{MPU}(g). \tag{9}$$

## 3.4 TEXT MULTISCALING

The proposed Multiscale PU Loss expects training texts of highly variant lengths, but training sets may contain lengthy paragraphs only. Therefore, we introduce Text Multiscaling Module that generates a variety of short texts to exert the potential of the length-sensitive Multiscale PU loss. We propose random deletion at sentence scale as a solution. Text Multiscaling module consists of 3 steps: first, a complete training text is first tokenized into $n$ sentences, denoted as sentence array $C$; then the sentences are independently and randomly masked based on a sentence-wise mask probability $p_{sent}$. In probabilistic terms, each sentence is decided by an independent Bernoulli trial in the sample space $\{0, 1\}$. In the sample space, 0 means the sentence is discarded and 1 stands for the sentence is maintained. Finally, all sentences are merged again for the multiscaled training text $c_{mul}$.

Mathematically, with $\odot$ stands for the element-wise Hadamard product, the above process could be concluded as:

$$c_{mul} = C \odot M, \quad \text{where } M \sim \text{Bernoulli}^n(1 - p_{sent}). \tag{10}$$

The proposed Text Multiscaling module is a one-to-one mapping from $C \to c_{mul}$; we are not generating more training samples, but substituting the original sample for fair comparison in experiments. Notably, it is probable that multiscale could leave the original text intact, or only one sentence is left. The relative sequence of remaining sentences is maintained to avoid breaking excess logical relations between sentences. Multiscaled texts automatically inherit class labels of their original text. The concern for attribution change due to length reduction is to be addressed by the use of Multiscale PU Loss.

Though random deletion is also applied in Easy Data Augmentation (EDA) (Wei & Zou, 2019), our method is different from theirs in two aspects. Firstly, our method is focused on multiscaling, while word-level random deletion proposed by EDA has limited effect in generating texts of various lengths. Secondly, EDA could break semantic meanings in sentences: deletion of keywords could change the class of a sentence; while a more integrated, sentence-level deletion reduces the chance of class property change.

## 4 EXPERIMENTS

### 4.1 SETTING OVERVIEW

**Datasets.** We choose TweepFake (Fagni et al., 2020) and HC3 (Guo et al., 2023) as benchmarks for our experiments. TweepFake (Fagni et al., 2020) is a dataset of tweets for AI-generated microblog detection. Since latest LLMs have completely reshaped the task of AI text detection, we also adopt HC3 (Guo et al., 2023), which is an up-to-date ChatGPT text detection dataset including both English and Chinese. Additionally, HC3 has short-text benchmarks: HC3-English-Sent and HC3-Chinese-Sent. We use these datasets to demonstrate the effectiveness of our method.

The length statistics in Table 2 show the distribution similarity of English short-text benchmarks, *i.e.* TweepFake (that consists of tweets) and HC3-En-Sent. We conclude from the statistics that the adopted HC3 short-text benchmark could simulate the fragmented language environment (*e.g.* Twitter) on mobile apps. Detector evaluation on these short-text benchmarks could reflect their real-world detection capabilities in smartphone-related scenarios.

| Benchmark | Mean | Std | Q1 | Q2 | Q3 |
|---|---|---|---|---|---|
| TweepFake (Fagni et al., 2020) | 24.82 | 15.19 | 13 | 21 | 34 |
| HC3-En-Sent (Guo et al., 2023) | 24.98 | 15.47 | 15 | 22 | 31 |

Table 2: Token length statistics of short-text benchmarks. HC3-English-Sent has a similar length distribution as TweepFake. These short-text benchmarks could simulate languages that we encounter in Instant Messaging and Microblogging Apps, like Twitter.

**Detectors.** BERT (Devlin et al., 2018) and RoBERTa (Liu et al., 2019) are adopted to apply our MPU method, due to their popularity and supreme performance in previous AI text detection works (Solaiman et al., 2019; Fagni et al., 2020; Liu et al., 2022; Guo et al., 2023). Training-agnostic detection algorithms are excluded from our consideration.

### 4.2 TWEEPFAKE DETECTION RESULTS

In TweepFake experiments, we follow Kumarage et al. (2023) for our training settings. Kumarage et al. (2023) is one of the latest works on AI-generated text detection, and it claims outstanding performance on short-text detection. We strictly follow the original training strategy in Kumarage et al. (2023): the model is trained with the AdamW optimizer at batchsize 16 and learning rate $1e-5$.

TweepFake mainly consists of short tweets. we inspect the dataset and find that a vast majority of texts are single or a handful of sentences. Hence, we refrain from using Text Multiscaling that

| Method | Acc. |
|---|---|
| BERT-Finetuned (Devlin et al., 2018) | 89.1 |
| RoBERTa-Finetuned (Liu et al., 2019) | 89.6 |
| RoBERTa-Stylo (Kumarage et al., 2023) | 91.1 |
| RoBERTa-MPU (Ours) | **91.4** |

Table 3: Experiments on short-text dataset TweepFake (Fagni et al., 2020).

randomly delete sentences for TweepFake datasets; rather, we directly apply Multiscale PU loss during training. As shown in Table 3, the experiment result of the proposed MPU is promising: it greatly improves the performance of finetuned RoBERTa, and its performance outcompetes the latest TweepFake baseline RoBERTa-Stylo (Kumarage et al., 2023) that requires an additional module for stylometric feature extraction during finetuning.

## 4.3 HC3-English Detection Results

| Method (F1 scores) | HC3-En-Full | HC3-En-Sent |
|---|---|---|
| GLTR (Gehrmann et al., 2019) | 96.52 | 40.19 |
| PPL (Guo et al., 2023) | 95.20 | 62.04 |
| OpenAI (OpenAI, 2023b) | 91.00 | 69.27 |
| DetectGPT (Mitchell et al., 2023) | 87.39 | 63.32 |
| BERT-Finetuned (Devlin et al., 2018) | 97.62±0.91 | 57.65±15.45 |
| RoBERTa-Finetuned (Liu et al., 2019) | 97.42±0.92 | 58.60±10.53 |
| RoBERTa-Stylo (Kumarage et al., 2023) | 96.48 | 81.46 |
| BERT-MPU (Ours) | **98.60**±0.52 | 79.76±3.07 |
| RoBERTa-MPU (Ours) | 98.40±0.31 | **85.31**±1.80 |

Table 4: Comparison with English AI-generated text detection baselines on HC3 Guo et al. (2023). Most baselines perform poorly on short texts (*i.e.* HC3-En-Sent); in contrast, our method improves short-text detection greatly.

We also experiment our method on ChatGPT corpora that are much harder to detect. In the ChatGPT text detection experiments, we follow the setting of HC3 (Guo et al., 2023) to test the performance of our method. HC3 (Guo et al., 2023) is a dataset targeted at ChatGPT text detection. All texts are reduced into shorter texts for a sentence-level variant. We apply the MPU framework on the full-scale dataset of HC3 (Guo et al., 2023).

Several baseline detectors are chosen to demonstrate the outstanding detection performance of our MPU method. These baselines are open-source and replicable. Among these baselines, GLTR (Gehrmann et al., 2019), PPL (Guo et al., 2023), and DetectGPT (Mitchell et al., 2023) are zero-shot methods that do not require further training: they rely on the likelihood outputs of a pretrained language model. The OpenAI Detector (OpenAI, 2023b) is a RoBERTa detector finetuned on OpenAI's GPT-2 (Radford et al., 2019) corpora. RoBERTa-Stylo Kumarage et al. (2023) is one of the latest detection baseline targeted for short texts. BERT-Finetuned and RoBERTa-Finetuned are language models plainly finetuned on HC3 (Guo et al., 2023), following the official setting; while BERT-MPU and RoBERTa-MPU are language models trained on HC3 (Guo et al., 2023) via the proposed MPU method.

It could be observed from Table 4 that most existing methods perform poorly on short texts. The statistics verify our previous claim that the detection of shorter texts is a difficult problem. Specifically, finetuned BERT and RoBERTa are good at detecting long, full-level texts, but they fail to filter out shorter AI-generated texts. On the contrary, our MPU method could greatly improve short-text performances and boost long AI-generated text detection as well. We will further investigate the effect of solitary MPU components in Sec. 4.5.

| Method | HC3-Ch-Full | HC3-Ch-Sent |
|---|---|---|
| GLTR (Gehrmann et al., 2019) | 87.40 | 49.94 |
| RoBERTa-Finetuned (Liu et al., 2019) | 96.28±3.42 | 83.07±6.85 |
| RoBERTa-MPU (Ours) | **97.42**±0.24 | **89.37**±1.94 |

Table 5: Comparison with Chinese AI-generated text detection baselines. Our method is also proved effective on Chinese corpora.

## 4.4 HC3-Chinese Detection Results

To verify the generality of the proposed MPU method in other languages, we also compare our method with baselines on Chinese AI text detection benchmark HC3-Chinese (Guo et al., 2023). Following Guo et al. (2023), we use chinese-roberta-wwm-ext (Cui et al., 2020) as the pretrained language model. The results are shown in Table 5. Our method could still outcompete other methods by large margins in terms of short-text detection, reaching an F1 score of 89.37 on HC3-Chinese-Sent.

## 4.5 Ablations

**Harmful Short Texts.** We elaborate in Section 3.1 that short texts could manifest a partially unlabeled property, which impacts the normal training process of the detector. To demonstrate that short texts are indeed harmful for training, we design an experiment based on the HC3-English dataset Guo et al. (2023) as follows: when the detector encounters a short training text during training, the training text is omitted from backward operations. Other settings are identical to Section 4.3. As shown in Table 6, finetuning without short texts demonstrates better performance compared with plain finetuning. This reveals that short sentences are harmful to detector training due to their partially unlabeled properties. Hence, PU frameworks need to be leveraged to address this issue.

| Method | HC3-En-Full | HC3-En-Sent |
|---|---|---|
| Finetuning with all texts | 97.42 ± 0.92 | 58.60 ± 10.53 |
| Finetuning without short sentences | **98.19** ± 0.66 | **62.42** ± 5.60 |

Table 6: Performance comparison between the detector finetuned with all texts and detector finetuned without short texts.

| Measures | | HC3-English | | HC3-Chinese | |
|---|---|---|---|---|---|
| Text Mul. | MPU loss | Full | Sent | Full | Sent |
| ✗ | ✗ | 97.42±0.92 | 58.60±10.53 | 96.28±3.42 | 83.07±6.85 |
| ✓ | ✗ | 96.42±2.27 | 82.76±2.76 | 95.89±4.18 | 84.79±5.94 |
| ✗ | ✓ | 97.48±2.41 | 45.30±8.78 | 96.87±0.89 | 83.46±5.78 |
| ✓ | ✓ | **98.40**±0.31 | **85.31**±1.80 | **97.42**±0.24 | **89.37**±1.94 |

Table 7: F1 scores of Finetuned RoBERTa on ChatGPT benchmark HC3. "Full" and "Sent" stands for model validated on long-text and short-text benchmarks, respectively.

**Framework Components.** We perform ablations on the solitary effects of Text Multiscaling and Multiscale PU loss.

From Table 7, it is firm that the addition of Text Multiscaling to training corpus greatly improves performance on sentence-level corpus detection as expected. Unfortunately, the detector's capability on full corpus decays. This performance drop is attributed to the unreasonable label assignment to short corpus from random sentence deletion: the generated short corpora automatically inherit labels from their full-level predecessors in Text Multiscaling Module, neglecting "unlabeled" properties as introduced in Sec. 3.1. The addition of MPU loss reverses full-level corpus detection performance drop and boosts short-text performance as well. Solitary addition of MPU loss only would have little help for detection performance for lack of short texts.

**MPU Loss.** We further investigate MPU loss configurations on ChatGPT text detection benchmark HC3-English (Guo et al., 2023).

The performance of Multiscale PU loss is evaluated against ordinary PU loss that disregards changes in sentence lengths, as shown in Table 8. Multiscale PU loss is sensitive to training corpora of various lengths and thus is more performant compared with its ordinary counterpart.

| PU type | Full | Sent |
|---|---|---|
| Ordinary | 97.05±2.15 | 83.53±3.14 |
| **Multiscale** | **98.40**±0.31 | **85.31**±1.80 |

Table 8: Performance comparison between ordinary PU loss and the proposed Multiscale PU loss.

Introduced in the abstract recurrent detection model (Sec. 3.3), token-wise prior $p$ estimates the probability of a token being highly characteristic as human-spoken. As shown in Table 9, we carefully tune $p$ and found that the best performance is reached at $p = 0.2$, which is small as we expect.

| $\gamma$ | Full | Sent | $p$ | Full | Sent | $p_{sent}$ | Full | Sent |
|---|---|---|---|---|---|---|---|---|
| 0 | 96.42±2.27 | 82.76±2.76 | 0.1 | 96.29±1.31 | **86.06**±1.97 | 0 | 97.48±2.41 | 45.30±8.78 |
| 0.2 | 96.52±0.38 | 83.94±4.07 | **0.2** | **98.40**±0.31 | 85.31±1.80 | 0.1 | 97.73±1.42 | 76.84±7.93 |
| **0.4** | **98.40**±0.31 | 85.31±1.80 | 0.3 | 96.81±1.70 | 84.17±2.78 | **0.25** | **98.40**±0.31 | 85.31±1.80 |
| 0.6 | 97.42±0.13 | **85.78**±1.19 | 0.4 | 97.44±1.06 | 82.88±3.32 | 0.4 | 97.45±1.34 | **87.11**±1.41 |
| 0.8 | 96.90±1.49 | 84.54±2.09 | | | | | | |

Table 9: Ablation experiment results on hyperparameters: loss proportion $\gamma$, the estimated probability of a token being clear-human $p$, and sentence mask probability $p_{sent}$.

We also carefully adjust the affine weight hyperparameter for PU loss $\gamma$, as shown in Table 9. As the affine weight $\gamma$ for PU loss gradually increases, the full-level corpus detection performance reaches the peak at $\gamma = 0.4$ and then drops, while the sentence-level performance reaches its peak at $\gamma = 0.6$. From a comprehensive perspective, the best overall performance is reached at $\gamma = 0.4$ where both performances on full and sentence-level corpus are satisfactory. The climb-and-drop trend reveals that short machine-generated sentences are not completely unlabeled; short-text classification should be viewed as a partial PU problem rather than a complete PU problem.

Further, we test the advantage of the non-negative risk estimator in the nnPU loss (Kiryo et al., 2017) against uPU loss (Du Plessis et al., 2014), as introduced in Sec. 3.2. The results are shown in Table 10.

| Loss type | Full | Sent |
|---|---|---|
| Unbiased PU (Du Plessis et al., 2014) | 97.90±0.25 | 84.87±1.28 |
| **Non-negative PU** (Kiryo et al., 2017) | **98.40**±0.31 | **85.31**±1.80 |

Table 10: Performance comparison between ordinary PU loss and the proposed Multiscale PU loss.

**Text Multiscaling.** As introduced in Sec. 3.4, we randomly mask sentences of the training set at probability $p_{sent}$ for multiscale text augmentation. We investigate on tuning $p_{sent}$ for the optimal value. The statistics are shown in Table 9. When $p_{sent}$ is set at 0.25, the test performance on both full and sentence level corpus are satisfactory; when $p_{sent}$ is set too high, sentence-level detection performance is enhanced, but full-level performance is negatively impacted because the full-scale training texts are overly damaged.

## 5    CONCLUSION

This paper proposes a Multiscale Positve-Unlabeled (MPU) framework for AI-generated text detection. We look at the iffy attribution of short AI-generated corpus, and model AI text detection as a partial PU problem. MPU loss and Text Multiscaling Module are to augment detectors' discriminative ability on short corpus.

ETHICS & REPRODUCIBILITY STATEMENT

This paper proposes a training method for AI-generated text detectors. Despite outstanding performance on multiscale texts, chances are that the detectors output the wrong attribution of a certain piece of text. This may cause ethical issues when the detector is used for detecting plagarism, fake news, et cetera. Hence, we strongly recommend that results from the detector could only serve as a reference in actual applications.

Experiments are reproducible. We have attached complete training settings in the Appendix; we also fix random seeds in our codes for the ease of replication. All details are in Appendix E.

ACKNOWLEDGEMENT

This work is supported by National Key R&D Program of China under Grant No.2022ZD0160300 and National Natural Science Foundation of China under Grant No.62276007. We gratefully acknowledge the support of MindSpore, CANN and Ascend AI Processor used for this research.

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

APPENDIX

## A  PU Loss Derivation

PU losses are derived from the canonical binary classification framework. In the standard supervised binary classification (or Positive-Negative classification, abbreviated as PN), let $\pi := p\,(Y = +1) = \frac{n_P}{n_P+n_N}$ be the prior probability of the positive class, $g : \mathbb{R}^d \to \mathbb{R}$ be an arbitrary decision function (in our case, the detector model) and $L$ be the loss function. The risk of $g$ is defined as the expectation of loss:

$$
\begin{aligned}
R(g) :=& \mathbb{E}_{(X,Y)\sim p(x,y)}[L(g(X), Y)] \\
=& \pi\mathbb{E}_p[L(g(X), +1)] + (1 - \pi)\mathbb{E}_n[L(g(X), -1)] \\
=& \pi R_P(g, +1) + (1 - \pi)R_N(g, -1).
\end{aligned}
\tag{11}
$$

In canonical PN learning, $R(g)$ can be approximated directly by losses calculated from training data as follows:

$$
\hat{R}_{PN}(g) = \pi\hat{R}_P(g, +1) + (1 - \pi)\hat{R}_N(g, -1),
\tag{12}
$$

where $\hat{R}_P(g, +1) := \frac{1}{n_P}\sum_{i=1}^{n_P} L(g(x_i^P), +1)$ and $\hat{R}_N(g, -1) := \frac{1}{n_N}\sum_{i=1}^{n_N} L(g(x_i^N), -1)$ are estimations of the positive and negative risk, respectively.

In the PU framework, $\hat{R}_N(g, -1)$ cannot be approximated directly via negtive samples. Alternatively, some works (Du Plessis et al., 2014; Plessis et al., 2015) perform indirect approximation as follows: defining $p_P(x) := p(x|Y = +1)$ and $p_N(x) := p(x|Y = -1)$, since

$$
(1 - \pi)p_N(x) = p(x) - \pi p_P(x),
\tag{13}
$$

the negative risk part (which is an expectation) is obtained as

$$
(1 - \pi)R_N(g, -1) = R_U(g, -1) - \pi R_P(g, -1),
\tag{14}
$$

and $R(g)$ can be approximated indirectly as

$$
\hat{R}_{uPU}(g) = \pi\hat{R}_P(g, +1) - \pi\hat{R}_P(g, -1) + \hat{R}_U(g, -1),
\tag{15}
$$

where $\hat{R}_P(g, -1) := \frac{1}{n_P}\sum_{i=1}^{n_P} L(g(x_i^P), -1)$ and $\hat{R}_U(g, -1) := \frac{1}{n_U}\sum_{i=1}^{n_U} L(g(x_i^U), -1)$ are estimations calculated from positive and unlabeled training samples. Eq. 15 is defined as the unbiased PU (uPU) loss (Du Plessis et al., 2014).

## B  Estimation Details of Confidence Expectation

**The transition matrix** Given positive probability $p$ of a single token, we express state transition as a band matrix $\mathbf{P}$. An example matrix form of $\mathbf{P}$ is listed as follows:

$$
\begin{bmatrix}
1-p & p & 0 & 0 & \dots & 0 & 0 & 0 \\
1-p & 0 & p & 0 & \dots & 0 & 0 & 0 \\
0 & 1-p & 0 & p & \dots & 0 & 0 & 0 \\
 & & & & \dots & & & \\
 & & & & \dots & & & \\
 & & & & \dots & & & \\
0 & 0 & 0 & 0 & \dots & 1-p & 0 & p \\
0 & 0 & 0 & 0 & \dots & 0 & 1-p & p
\end{bmatrix}
$$

**Demonstration of $\tilde{\pi}$ increment with respect to lengths** We try to mathematically demonstrate that prior $\tilde{\pi}$ increases with length $l$. The initial state $\sigma_0$ is one-hot, so the prior $\tilde{\pi}(l)$ with respect to $l$ could be written as:

$$
\tilde{\pi}(l) = E\left[\Delta(S_l)\right] = \sigma_0\mathbf{P}^l\alpha^T = \mathbf{P}[n, :]\mathbf{P}^{l-1}\alpha^T,
\tag{16}
$$

where $\mathbf{P}[n,:]$ represents the last row of transition matrix $\mathbf{P}$. To demonstrate $\tilde{\pi}$ increases with $l$, we alternatively demonstrate $\tilde{\pi}(l+1) - \tilde{\pi}(l) = E\left[\Delta(S_{l+1})\right] - E\left[\Delta(S_l)\right]$ is positive.

However, sizes of states and transition matrices are different for corpora of different lengths. We use a subscript to indicate this difference. For instance, sequence vector $\alpha_l := [i/l]_{i=0}^{l}$ indicates all possible confidences in a sorted sequence; $\mathbf{P}_l$ indicates the transition matrix $\mathbf{P}$ of size $(l+1) \times (l+1)$. Then:

$$E\left[\Delta(S_{l+1})\right] - E\left[\Delta(S_l)\right] = \mathbf{P}_{l+1}[n,:]\mathbf{P}_{l+1}^{l-1}\mathbf{P}_{l+1}\alpha_{l+1}^T - \mathbf{P}_l[n,:]\mathbf{P}_l^{l-1}\alpha_l^T \tag{17}$$

Interestingly, we could leverage unique features of the sparse band matrix $\mathbf{P}$. First, obviously $\mathbf{P}_{l+1}[n,:] = [0; \mathbf{P}_l[n,:]]$. Further, if we compare

$$M := \mathbf{P}_{l+1}[n,:]\mathbf{P}_{l+1}^{l-1} \in \mathbb{R}^{l+2} \quad \text{and} \quad K := \mathbf{P}_l[n,:]\mathbf{P}_l^{l-1} \in \mathbb{R}^{l+1},$$

we would discover that $M = [0; K]$, namely, array $M$ is array $K$ prepended by a zero. (The physical meaning of $M$ and $K$ is the last line of matrix $\mathbf{P}_{l+1}^l$ and $\mathbf{P}_l^l$, respectively.) Based on this discovery, we could simplify Eq. 17:

$$E\left[\Delta(S_{l+1})\right] - E\left[\Delta(S_l)\right] = [0; K]\mathbf{P}_{l+1}\alpha_{l+1}^T - K\alpha_l^T \tag{18}$$

Then we look at the concrete form of $[0; K]\mathbf{P}_{l+1}$. For simplicity, we denote the $n^{th}$ element of $K$ as $k_n$:

| Count | 0 | 1 | 2 | ... | $n$ | $n+1$ |
|---|---|---|---|---|---|---|
| $[0; K]$ | 0 | $k_0$ | $k_1$ | ... | $k_{n-1}$ | $k_n$ |
| $[0; K]\mathbf{P}_{l+1}$ | $(1-p)k_0$ | $(1-p)k_1$ | $pk_0 + (1-p)k_2$ | ... | $pk_{n-2} + (1-p)k_n$ | $pk_{n-1} + pk_n$ |

Based on the table above, we could derive the relations between $E\left[\Delta(S_{l+1})\right]$ and $E\left[\Delta(S_l)\right]$:

$$
\begin{aligned}
E\left[\Delta(S_{l+1})\right] - E\left[\Delta(S_l)\right] &= \frac{\sum_{n=0}^{l} nk_n}{l+1} + \frac{2p}{l+1} - \frac{k_l p}{l+1} - \frac{\sum_{n=0}^{l} nk_n}{l} \\
&= -\frac{\sum_{n=0}^{l} nk_n}{(l+1)l} + \frac{2p - k_l p}{l+1} \\
&= -\frac{E\left[\Delta(S_l)\right]}{l+1} + \frac{2p - k_l p}{l+1},
\end{aligned}
\tag{19}
$$

which means that

$$E\left[\Delta(S_{l+1})\right] = \frac{l}{l+1} E\left[\Delta(S_l)\right] + \frac{2p - k_l p}{l+1}, \tag{20}$$

As long as we view $\{l \times E\left[\Delta(S_l)\right]\}$ as a sequence of corpus length $l$ starting from $1 \times E\left[\Delta(S_1)\right] = p$, we could solve $E\left[\Delta(S_l)\right]$ for $l > 1$:

$$E\left[\Delta(S_l)\right] = \frac{(2l-1)p - p\sum_{n=1}^{l-1} k_{n,n}}{l} = 2p - \frac{p}{l}(1 + \sum_{n=1}^{l-1} k_{n,n}), \tag{21}$$

where $k_{n,n}$ is the probability of the abstract recurrent model outputting positive confidence 1 for a corpus of length $n$. However, we encounter the difficulty that the analytic solution to $k_{n,n}$ is not easily solvable; we only know that $k_{n,n}$ is a probability bounded in $(0, 1)$. We inspect $k_{n,n}$ for relatively small $p$ and found that $k_{n,n}$ quickly converges to 0. This process is demonstrated by Figure 1, where $k_{n,n}$ decays in an approximately exponential manner to infinitesimally small values (which decays much faster than reciprocals, i.e. $1/l$). As a result, prior $\tilde{\pi}$ keeps increasing

as $l$ increases, and converges to $2p$. Figure 1 (Right) confirms the convergence derived in Eq. 21.

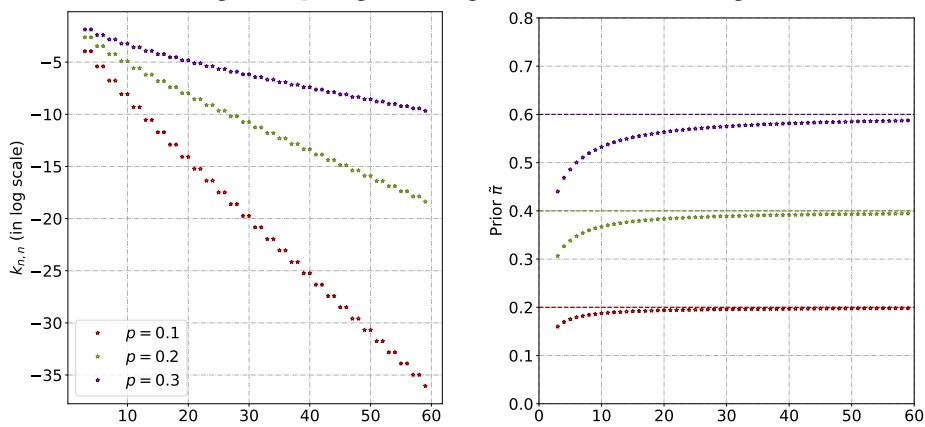

Figure 1: **Left:** $k_{n,n}$ (in log scale) with respect to corpus length $l$. **Right:** $\tilde{\pi}$ with respect to corpus length $l$.

## C   PROPOSAL OF IMPOSING SPACE CLEANING ON THE HC3-ENGLISH BENCHMARK

We use the HC3 (Guo et al., 2023) benchmark for ChatGPT corpus detection experiments. However, we inspected HC3 corpora and discovered that the corpora are flawed: human corpora have additional spaces before punctuations, while corpora from AI do not have this feature. The extra spacing could directly impact the input to detectors. We list several examples below, demonstrating the obvious difference between Human and ChatGPT corpora in the HC3 benchmark (Guo et al., 2023):

```
# labeled as Human
corpus = 'Basically there are many categories of " Best Seller " .'
input_ids = [0, 34480, 89, 32, 171, 6363, 9, 22, 2700, 44795, 22, 479, 2]

corpus = 'Same thing for best sellers .'
input_ids = [0, 42271, 631, 13, 275, 12649, 479, 2]

corpus = 'Also , IIRC the rankings change every week or something like
    that .'
input_ids = [0, 22412, 2156, 3082, 5199, 5, 8359, 464, 358, 186, 50, 402,
    101, 14, 479, 2]

# labeled as ChatGPT
corpus = 'It is generally not acceptable or ethical to advocate for or
    condone the assassination of any individual , regardless of their
    actions or beliefs.'
input_ids = [0, 243, 16, 3489, 45, 9796, 50, 13557, 7, 7156, 13, 50,
    35005, 5, 16351, 9, 143, 1736, 6, 6069, 9, 49, 2163, 50, 9734, 4, 2]

corpus = 'There are also practical considerations at play in this
    situation .'
input_ids = [0, 970, 32, 67, 7708, 19199, 23, 310, 11, 42, 1068, 4, 2]

corpus = 'It can also lead to further conflict and instability in the
    region .'
input_ids = [0, 243, 64, 67, 483, 7, 617, 3050, 8, 16826, 11, 5, 976, 4,
    2]
```

In the examples, we show original corpus as well as their token ids after being processed by the RoBERTa-base tokenizer. Most human corpora have an unexpected 479 token (standing for " .", i.e. a space and a period), while ChatGPT corpora does not manifest this feature.

Hence, the detector could judge the attribution of a certain corpus simply by detecting these spacing mistakes. Embarrasingly, if we use the logical judgement of whether token id 479 is contained in the sequence to detect human corpora, the F1 score would reach $82.12\%$ on sentence-level test corpora of the HC3 benchmark. The performance of such a simple logic is even better than the officially reported performance ($81.89\%$) of finetuned RoBERTa-base (Guo et al., 2023). Above all, we strongly recommend later works that involve the HC3 benchmark to remove unnecessary spaces before punctuations. We will opensource the code simple cleaning helper function that removes unnecessary spaces.

## D    BASELINE REPLICATIONS

### D.1    DETECTGPT

DetectGPT (Mitchell et al., 2023) is a latest open-sourced AI corpus detection baseline, but the original paper did not report its performance on latest LLM texts. Hence, we replicate DetectGPT on the HC3-English (Guo et al., 2023) ChatGPT corpus dataset, and compare it with our MPU method. The experiment results are shown in Table 4, where our MPU method outcompetes DetectGPT by large margins. There is still a visible gap between latest training-agnostic methods (e.g. DetectGPT) and finetuned language models on ChatGPT corpora.

We also provide some detailed procedures to tailor DetectGPT for the HC3 benchmark: 1. Full-scale HC3 corpora are always too long to perturb. Therefore, we truncate corpora as long as they raise perturbation errors, following recommendations from authors of DetectGPT. 2. We use 100 perturbations for full-scale HC3 corpora (following DetectGPT (Mitchell et al., 2023)), but we use 10 perturbations for sentence-level HC3 because there are too many corpora. It also reflects that DetectGPT is not very efficient for large-scale corpora compared to language model detectors, because it requires tens of model runs for a single corpus. 3. DetectGPT uses AUROC as the classification metric; however, this metric is not applicable to finetuned language models that output probabilities for respective classes. Hence, given confidences of all corpora outputted from DetectGPT, we choose 1000 equally-spaced threshold between max and min values, and maintain the threshold with the largest F1 score. Notably, this will provide an upperbound for the performance of DetectGPT, as in real applications the threshold is pre-set; scanning for the best threshold on test sets is strictly prohibited.

### D.2    GLTR, PPL, & OPENAI

These methods have already been open-sourced on HuggingFace. We directly input all texts in the testset to these baseline methods and measure their performances.

We have found an inconsistency in comparison to reported values while replicating GLTR (Gehrmann et al., 2019) and RoBERTa-Finetuned (Cui et al., 2020) on the HC3-Chinese (Guo et al., 2023) benchmark, shown in Table 11. This inconsistency is tolerable and won't affect our final conclusion.

| Method | Full | Sent |
|---|---|---|
| GLTR (Reported by Guo et al. (2023)) | 89.61 | 44.02 |
| GLTR (Replicated) | 87.40 | 49.94 |
| RoBERTa-Finetuned (Reported by Guo et al. (2023)) | 98.79 | 83.64 |
| RoBERTa (Replicated) | 96.28±3.42 | 83.07±6.85 |

Table 11: Our replication of HC3-Chinese Guo et al. (2023) baselines compared with reported values.

## E    REPLICATION DETAILS

Following the training setting of Kumarage et al. (2023), we use batchsize 16, learning rate $1e-5$ for TweepFake; following the setting of Guo et al. (2023), we use batchsize 32, learning rate $5e-5$ for HC3. AdamW optimizors are adopted. Selected benchmarks are publicly accessible online.

We use a single Nvidia Tesla V100 as the device for experiments. A single epoch of training costs around 30 minutes. We replicate all experiments three times to avoid fluctuation, using seed=0,1,2. The codes are opensourced at GitHub and Gitee.

