# OpenReview forum: "Multiscale Positive-Unlabeled Detection of AI-Generated Texts"
_ICLR.cc/2024/Conference — ICLR 2024 spotlight_

### Official Review · Reviewer_vhmU · 2023-10-30

**Soundness:** 3 good
**Presentation:** 3 good
**Contribution:** 4 excellent
**Rating:** 6
**Confidence:** 4

**Summary:**

This paper models the task of AI-generated text detection as a partial Positive-Unlabeled (PU) problem and formulates the Multiscale Positive-Unlabeled (MPU) training framework to address the challenging task of short text detection without sacrificing long texts. In the PU context, this paper proposes length-sensitive Multiscale PU Loss, where a recurrent model in abstraction is used to estimate positive priors of scale-variant corpora. Besides, this paper introduces a Text Multiscaling module to enrich training corpora. Experimental results can demonstrate the effectiveness of the proposed method.

**Strengths:**

- It is interesting and meaningful to investigate the task of AI-generated text detection.
- It is a novel perspective to address the task of AI-generated text detection by taking it as a partial Positive-Unlabeled (PU) problem.
- A novel Multiscale Positive-Unlabeled (MPU) training framework is proposed to address the challenging task of short text detection without sacrificing long texts.
- A Text Multiscaling module is also introduced to enrich training corpora.
- Experimental results are extensive, which can demonstrate the effectiveness of the proposed method.
- Ablation studies also support the proposed method.

**Weaknesses:**

- The writing of this paper should be further polished. For example, the double quotation mark was mistakenly used in the paper. For the typeset, there are too many tables that record the experimental results in the Experiment section. Maybe some tables can merged into a single one. It would be better if some tables could be transformed into figures. There are some words that are on the 10th page, but the requirement is that the main content cannot exceed 9 pages.
- The review of related work can be further improved. Section 2.2 talks too much about PU learning. I think it is not necessary to provide the unbiased PU loss and the non-negative PU loss in such a detailed manner. In Section 1 and Section 2, some parts could be arranged into a single Related Work section.

**Questions:**

Please check the weaknesses.

---

> ### Author Response · Authors · 2023-11-20
> **Response to Reviewer vhmU**
>
> Dear reviewer vhmU,
>
> Thank you very much for your review. Here are our responses:
>
> *W1: The writing of this paper should be further polished.*
>
> *A1:* Sorry for the writing problems in the paper. We have further polished our paper according to your advice as follows: firstly, we corrected double quotation mark mistakes. Secondly, we merge small ablations tables into a single large one to make the paper layout neater. Thirdly, though having "Ethics & Repoducibility Statement" on the 10th page is allowed, we think there is much redundancy in the paper and we reduce unnecessary parts to put everything except reference within 9 pages.
>
> *W2: The review of related work can be further improved.*
>
> *A2:* Thanks for your advice. We have revised that part according to your advice.
>
> Sincerely,
>
> Authors

---

### Official Review · Reviewer_PQsn · 2023-10-31

**Soundness:** 3 good
**Presentation:** 2 fair
**Contribution:** 3 good
**Rating:** 6
**Confidence:** 5

**Summary:**

This paper is motivated by the challenge of classifying short texts as either human or machine generated, i.e. fake text detection of short documents. The paper suggests treating short documents from machines as *unlabeled*, under the premise that such texts are in fact indistinguishable from human-written ones in some cases. From this assumption, the "hammer" of positive-unlabeled (PU) learning is applied to the problem. Specifically, a length-sensitive PU approach is proposed to account for the prior probability of positive-labeled data, which requires a "multi-scaling" data augmentation (sentence-level masking) to introduce a variety of sentence lengths at training time. On the TweepFake dataset, the approached approach outperforms a BERT-based classifier by 0.3 in terms of accuracy. On the more recent HC3 dataset, the proposed approach appears to outperform BERT-based classifiers by about 1% in accuracy for English and Chinese on full documents with larger improvements on sentence-level tasks (e.g., 85% vs 81% for a vanilla classifier for HC3-En-Sent)

**Strengths:**

* The paper addresses an increasingly important application, namely the automatic detection of machine-generated texts.
* The approach appears to outperforms BERT-based classifiers across two benchmarks, both on long texts and at sentence-level detection. The improvements on sentence-level tasks are notable and appear fairly consistent across datasets and languages.
* Ablations of the proposed approach are reported, showing the benefits of each component of the pipeline.

**Weaknesses:**

* The motivation for using a PU framework for fake text detection is not entirely clear. Conceptually, although certain short texts may be hard to distinguish as AI generated, pretending that they are unlabeled seems like an odd choice. Is there evidence that such shorts texts actually harm learning of a classifier using typical proper losses?

* If the benefit is actually prioritizing learning on examples with less irreducible uncertainty, might other frameworks besides PU be appropriate as baselines? (e.g., noise-aware losses)

* How does the use of the PU loss impact calibration? Uncertainty estimation is important for downstream use cases of AI text detectors.

* S2.3 is difficult to understand. There is a lot of complexity without sufficient motivation or intuition for what it is aiming to accomplish, or discussion of alternate (simpler) approaches.

* Eqns (12) -- (14) introduce a lot of unnecessary notation that is never re-used elsewhere in the paper.

EDIT: Dear authors, thank you for the detailed response. I have updated my review accordingly.

**Questions:**

* It's unclear why a recurrent model is necessary in S2.3. Would an attention-based model also work here?

---

> ### Author Response · Authors · 2023-11-20
> **Response to Reviewer PQsn**
>
> Dear reviewer PQsn,
>
> Thank you very much for your review. Here are our responses:
>
> *W1: The motivation for using a PU framework for fake text detection is not entirely clear: Is there evidence that such shorts texts actually harm learning of a classifier using typical proper losses?*
>
> *A1:* We have conducted experiments to prove that such short texts harm the learning of the AI text detector. During finetuning, we remove sentences if they are too short. We do not change any other settings, including training hyperparameters and the dataset. As shown in the table below, compared with finetuning with all sorts of texts, finetuning without short sentences could achieve better performance on full-scale long texts and shorter sentence-level texts. The results indicate that short texts are harmful in finetuning. However, there is still much room for improvement in terms of short text performance. Hence, we try to leverage these "harmful" short texts in the PU framework rather than remove them in finetuning.
>
>
> |   Method            |    HC3-English-Full         |    HC3-English-Sent   |
> | ---------------------------- | ----------------------- | ----------------------- |
> | Finetuning with all texts   | 97.42 $\pm$ 0.92 | 58.60 $\pm$ 10.53 |
> | Finetuning without short sentences              | 98.19 $\pm$ 0.66 | 62.42 $\pm$ 5.60 |
> | MPU (Ours)             | **98.40** $\pm$ 0.31 | **85.31** $\pm$ 1.80 |
>
> *W2: Might other frameworks (e.g., noise-aware losses) besides PU be appropriate as baselines?*
>
> *A2:* Thanks for your suggestions. We add experiments on several popular noise-aware losses. The adopted noise-aware losses are as follows: MAE [1], RCE [2], and GCE [3]. These experiments use the identical training setting as our MPU loss experiments. The results are shown in the table below, revealing that the proposed MPU framework gains the upper hand.
>
>
>
> |   Method            |    HC3-English-Full         |    HC3-English-Sent   |
> | ---------------------------- | ----------------------- | ----------------------- |
> | MAE [1]   | 97.28 $\pm$ 1.38 | 82.61 $\pm$ 4.64 |
> | RCE [2]   | 89.32 $\pm$ 13.58 | 68.17 $\pm$ 7.92 |
> | GCE [3]   | 97.63 $\pm$ 2.24 | 74.01 $\pm$ 7.92 |
> | MPU (Ours)             | **98.40** $\pm$ 0.31 | **85.31** $\pm$ 1.80 |
>
>
> *W3: How does the use of the PU loss impact calibration? Uncertainty estimation is important for downstream use cases of AI text detectors.*
>
> *A3:* We conduct experiments on the HC3-English benchmark and report the calibration measures: Expected Calibration Error (ECE) and Maximum Calibration Error (MCE) [4] on the full and sent datasets to reflect how calibration is impacted. It can be observed that MPU reduces error on both metrics.
>
> |                     | ECE (Full) | MCE (Full) | ECE (Sent) | MCE (Sent) |
> | ------------------- | ---------- | ---------- | ---------- | ---------- |
> | Ordinary Finetuning | 1.51       | 24.73      | 10.36      | 30.08      |
> | Finetuning with MPU | 0.47       | 20.79      | 6.51       | 22.02      |
>
>
>
> *W4: Section 2.3 is difficult to understand.*
>
> *A4:* Thank you very much for your advice. We have added more explanation about motivation in the latest revision.
>
> *W5: Too many unnecessary notations.*
>
> *A5:* We are sorry for the redundancy of notations. We have reduced them to make the equations concise.
>
>
> *Q1. Would an attention-based model also work here?*
>
> *A1:* We admit that attention is more powerful in language modeling compared to Recurrent Networks, as they jump out of the Markovian chain and establishes the dependency of one token with many other tokens. However, an attention-based might not work here in our scenario, because it is too complicated for our need of estimating the prior. We are estimating the general probability of a sample being positive without concrete samples. However, attention relies on concrete texts to establish the complex and precise dependency. Hence, we hold that an attention-based model might not work here.
>
> Sincerely,
>
> Authors
>
>
>
> References
>
> [1] Ghosh, Aritra, Himanshu Kumar, and P. Shanti Sastry. "Robust loss functions under label noise for deep neural networks." *Proceedings of the AAAI conference on artificial intelligence*. Vol. 31. No. 1. 2017.
>
> [2] Zhang, Zhilu, and Mert Sabuncu. "Generalized cross entropy loss for training deep neural networks with noisy labels." *Advances in neural information processing systems 31*. 2018.
>
> [3] Wang, Yisen, et al. "Symmetric cross entropy for robust learning with noisy labels." *Proceedings of the IEEE/CVF international conference on computer vision*. 2019.
>
> [4] Naeini, Mahdi Pakdaman, Gregory Cooper, and Milos Hauskrecht. "Obtaining well calibrated probabilities using bayesian binning." In *Proceedings of the AAAI conference on artificial intelligence*, vol. 29, no. 1. 2015.

---

### Official Review · Reviewer_scbq · 2023-11-01

**Soundness:** 3 good
**Presentation:** 4 excellent
**Contribution:** 3 good
**Rating:** 6
**Confidence:** 4

**Summary:**

It is difficult to classify shorter text as AI or human-generated text due to simple structure or phrasing. Motivated by this, the paper presents an innovative approach to classifying text as either AI or human-generated by framing it as a partially unlabeled (PU) task. The authors introduce a length-sensitive prior that complements the original PU loss function. They leverage a recurrent discriminator to determine the extent of unlabeled properties in each instance and introduce a multi-scale training strategy involving random sentence removal based on a Bernoulli distribution. The experimental results demonstrate that this approach yields better performance for short texts.

**Strengths:**

1. The paper is well-written and easy to follow.
2.  With the increasing popularity of LLMs, classifying AI or human-generated text is a crucial task. Short texts present a unique challenge due to their simplistic structure, making this research particularly relevant.
3. The motivation behind formulating AI-generated short-text classification as a partially unlabeled task is reasonable and aligns with the properties of classifying short texts.
4. The introduction of a PU loss function that takes text length into consideration is a novel and promising contribution to the field.

**Weaknesses:**

1. It's worth noting that some hyper-parameters require careful tuning for optimal performance. The authors should consider discussing the practical implications of tuning parameters like γ, especially when dealing with different large language models.
2. To enhance the paper's overall impact, it would be more convincing if the authors conducted an analysis of the performance of general recurrent language models. This analysis could include examining how well such models align with ground-truth labels and providing case studies to illustrate their effectiveness.

**Questions:**

1. In Table 4, it's intriguing to observe that BERT-MPU performs better than Roberta-MPU on full text but worse on short text. The authors could provide further insights or hypotheses as to why this performance gap exists. This could help readers better understand the nuances of the approach and its applicability in different scenarios.
2. It would be interesting to know whether the model's performance on classifying short texts would improve if the training dataset only consists of short texts.
3. Does the proposed method harm the performance of extremely long texts? For extremely long text framing classification as a PU task may not be a good option.

---

> ### Author Response · Authors · 2023-11-21
> **Response to Reviewer scbq**
>
> Dear reviewer scbq,
>
> Thank you very much for your review. Here are our responses:
>
> *W1: The authors should consider discussing the practical implications of tuning parameters like $\gamma$, especially when dealing with different large language models.*
>
> *A1:* Thanks for your advice. The magnitude of $\gamma$ implies the amount of "unlabeledness" in short AI texts. We hold that under identical training settings, $\gamma$ need to be set at higher values when large language models are more powerful. The reason is that LLMs that are more capable could generate sentences that are more alike to human. The "unlabeled" property is thus manifested more on short texts.
>
> *W2: It would be more convincing if the authors conducted an analysis of the performance of general recurrent language models. This analysis could include examining how well such models align with ground-truth labels and providing case studies to illustrate their effectiveness.*
>
> *A2:* Thanks for your advice of investigating the general recurrent models. We are afraid that we could not examine the models' alignment with ground-truth labels, because the attribution of short texts is a latent property. As explained in the paper, short texts are partially unlabeled. We could not know the precise attribution of a certain piece of text, and thus the label alignment analyses on the recurrent model could not be conducted. However, we could demonstrate the effectiveness of these general recurrent models via the detector's performance on benchmarks. As shown in the table below, we have compared the detector's performance with or without the recurrent model on the HC3-English benchmark in order to demonstrate the effectiveness of the General recurrent model.
>
> |                     | HC3-English Full | HC3-English Sent |
> | ------------------- | ---------- | ---------- |
> | w/o General Recurrent Language Models | 97.05 $\pm$ 2.15  | 83.53 $\pm$ 3.14 |
> | w/ General Recurrent Language Models | **98.40** $\pm$ 0.31  | **85.31** $\pm$ 1.80 |
>
> *Q1: Why does the BERT-MPU perform better than Roberta-MPU on full text but worse on short text?*
>
> *A1:* We hold that this gap is caused by the difference in difficulty of the two tasks, *i.e.* long text detection is much easier than short ext detection. Both BERT-MPU and RoBERTa-MPU reaches the performance supreme of full-level long texts; the rest of the minor cases that are falsely classified by both could be too difficult for existing detectors. In contrast, short text detection is a tricky task with more room for improvement. The gap of detection capability between BERT and RoBERTa is thus demonstrated.
>
> *Q2: Would the model's performance on classifying short texts improve if the training dataset only consists of short texts?*
>
> *A2:* Yes, MPU would also improve detectors that are trained on short texts only. In the TweepFake benchmark, all texts are short texts according to its token length statistics:
>
> | Benchmark | Mean  | Std   | Q1   | Q2   | Q3   |
> | --------- | ----- | ----- | ---- | ---- | ---- |
> | TweepFake | 24.82 | 15.19 | 13   | 21   | 34   |
>
> A performance improvement is also observed when the proposed MPU method is applied to RoBERTa according to the table below:
>
> | Method                           | Accuracy |
> | --------------------------------- | -------- |
> | RoBERTa (Ordinary Finetuning) | 89.6     |
> | RoBERTa-MPU (Ours)                | 91.4     |
>
> *Q3: Does the proposed method harm the performance of extremely long texts?*
>
> *A3:* We analyze the performance on extremely long texts, which is defined as sentences longer than token length 256. As shown in the table below, results demonstrate that MPU actually improves extremely long texts.
>
> | Method                           | F1 Score |
> | --------------------------------- | -------- |
> | RoBERTa-Finetuned | 97.16     |
> | RoBERTa-MPU (Ours)                | 99.77     |
>
>
> Sincerely,
>
> Authors

---

### Official Review · Reviewer_o5iQ · 2023-11-05

**Soundness:** 3 good
**Presentation:** 3 good
**Contribution:** 3 good
**Rating:** 8
**Confidence:** 2

**Summary:**

This paper addresses the challenge of detecting AI-generated text, which is particularly difficult for short texts. The authors acknowledge that some short texts generated by AI closely resemble human-generated ones and should be labeled as "unlabeled." To tackle this issue, the paper proposes a partial PU method designed for the detection of both long and short texts. Additionally, the authors introduce a corresponding loss function and employ a recurrent model to estimate the prior of P data. The experimental results demonstrate significant success.

**Strengths:**

1. The problem tackled in this paper is novel. The novelty of the problem addressed in this paper lies in the unique challenge of detecting AI-generated text, especially in the context of short texts. This is an important and emerging area of research with potential applications in various domains.

2. The proposed method is innovative, especially in its consideration of prior estimation dynamically. The experimental results are also promising.  The experimental results demonstrate the effectiveness of the proposed method, which further strengthens the validity of the approach.

3. The paper is well-written and presents the concepts clearly.

**Weaknesses:**

1. The organization of the paper could be improved. Section 2 contains a substantial amount of background information (Sections 2.1 and 2.2), and some content overlaps with Section 1. It is advisable to reorganize and eliminate redundancy for better flow and clarity.

2. There are some instances of misleading information in Section 2.2 regarding the estimation and expectation of PU learning. Specifically, the description above Eq1 poses a problem as \hat R represents an estimation rather than an expectation. Similarly, Eq 2 may hold for expectation but not necessarily for empirical estimation. Additionally, \tilde \pi should ideally be identical to \pi for the expected version of Eq 3 to be valid. These points should be clarified and corrected for accuracy.

**Questions:**

What specific factors contribute to the difficulty in detecting short text? The paper seems to suggest that the challenge lies in the resemblance between AI-generated short texts and human-generated ones. If this is the case, is it truly necessary to distinguish between them? Additionally, is it feasible to accurately differentiate between them?

Could you provide further insights into why \tilde \pi exhibits variation with respect to text length? It appears that it should reflect a characteristic of the dataset's distribution rather than the length of individual data points. Clarification on this point would enhance the understanding of the methodology.

---

> ### Author Response · Authors · 2023-11-20
> **Response to Reviewer o5iQ**
>
> Dear reviewer o5iQ,
>
> Thank you very much for your review. Here are our responses:
>
> *W1: The organization of the paper could be improved.*
>
> Thank you very much for your advice. According to your recommendations, we have re-organized the paper to reduce redundancy in the latest revision.
>
> *W2: There are some instances of misleading information in Section 2.2 regarding the estimation and expectation of PU learning.*
>
> Sorry for the misleading flaws in Section 2.2. We have corrected these mistakes in the latest revision (now in Appendix A).
>
> *Q1: What specific factors contribute to the difficulty in detecting short text? Is it feasible to accuurately differentiate between them?*
>
> The factor contributing to the difficulty in detecting short text lies in the reduced amount of information in contrast to longer texts. For longer texts, due to more information provided, they have prominent attribution to either AI or human. Hence, it is unsuitable two put both long and short text classification under the same PN classification framework.
>
> In spite of the difficulty, we still hold that most short texts have clear attribution. As explained in "Introduction", only extremely short, information-poor texts are indistinguishable. For ordinary short sentences (like Example 1 in Table 1), the division between human and AI texts is clear from factors like punctuations, emotions, and formality. Classsifying short text is feasible.
>
>
> *Q2: Could you provide further insights into why \tilde \pi exhibits variation with respect to text length?*
>
> In fact, we do agree that \tilde \pi is based on the distribution of the whole dataset in the PU framework. However, for our case with multiscale texts, the difficulty of detecting long or short texts varys. Hence, long texts and short texts have different distributions, and it could be unsuitable for all kinds texts sharing different distributions to share the same prior \tilde \pi.  In details, we hold that the distribution shift from long text to short text is a gradual process with respect to text lengths. In a differentiation perspective, texts of a certain length could be regarded as a small subset that features its own distribution and its own prior \pi. By setting prior estimation \tilde \pi as varying with respect to length, the gradual distribution change is considered. We have revised Section 2.3 (now Section 3.3) by adding further explanations.
>
> Sincerely,
>
> Authors

---

### Author Response · Authors · 2023-11-20
**Revision Summary**

Dear AC and reviewers,

We have revised the paper according to the reviews. The modifications are as follows:

1. Addition of Section "Related Work": we merged reviews of text detection methods from Section 2.1 (now 3.1), and reviews of PU methods in Section 2.2 (now 3.2).

2. Revision of Section 2.2 (now 3.2) "Preliminaries: PU Classification": we moved most derivation of the PU loss to the Appendix. In addition, we clarified the concept and notation of "expectation" and "estimation".

3. Clarification of Section 2.3 (now 3.3): we refuted the commonly used PU practice of constant priors in the multiscale text detection problem. Further, we explained the motivation of setting length-variant priors for easy understanding of this section.

4. Reduction of Section 2.4 (now 3.4): we reduced redundant notations and formulae to make the section concise.

5. Additional Ablation Experiments in Section 3.5 (now 4.5): We have added additional experiments to illustrate the harm of short texts during training.

6. Merger of tiny tables in Section 3.5 (now 4.5): We merged three tiny ablation experiment tables (Table 8,9,10) into one large table (Table 8).



P.S. We have marked revised parts red. For clarity, we mark the title rather than the entire body paragraph of sections that are newly added or significantly revised.

Sincerely,

Authors

---

### Meta-Review · Area_Chair_w9fE · 2023-11-30

**Metareview:**

This is a problem-driven methodology paper. It focused on an extremely important problem --- detection of AI-generated texts, more specifically, how to improve the detection performance of AI-generated *short* texts while not sacrificing the detection performance of AI-generated *long* texts. From the methodology point of view, it proposed a novel framework called *multi-scale positive-unlabeled learning* to handle the aforementioned problem. This paper is motivated and has solid contributions (including also sentence-wise random deletion in Sec 3.4). It has received consistently positive reviews, and thus we should definitely accept it for publication at ICLR 2024.

I went through all sections before the experiments (the revised version, not the submitted version) and found writing can still be improved. When I first saw Table 1, I didn't understand why Example 2 should be regarded as unlabeled and why it inspired you to regard Example 1 as partially unlabeled. Considering Example 2, I feel we should adopt "[0.5,0.5]" as the target of the cross-entropy loss because $p(y=+1|x)=p(y=-1|x)$ --- this uncertainty is intrinsic after the true (soft-)label is revealed, which is quite different from the uncertainty of being unlabeled before revealing the true label. Moreover, considering Example 1, it holds that $p(y=+1|x)$ and $p(y=-1|x)$ are very close but not equal to each other, so why this should be unlabeled? The data similar to Example 1 are the hard data that we must focus on during training, like support vectors for the support vector machines in the old days.

Then, I found the answers in the end of Sec 3.3, from Eq. (9) itself and its explanation: "As is emphasized, short machine-generated texts should be viewed as partially 'unlabeled' rather than entirely 'unlabeled'. Hence, we weight-sum the multiscale PU loss and the canonical PN classification loss ..." But where did the authors emphasize this point previously? I think the authors never explained what they meant by *partially* unlabeled until Eq. (9) even though this term was frequently used since the abstract! Indeed, the proposed MPU loss is supposed to be used together with the standard PN loss, which I think should be mentioned as early as in the abstract and then nicely explained with details in the introduction around Table 1. This is really the most confusing part of the paper.

I haven't detected other major issues but there seem to be several minor issues, mainly in writing. For example, "where texts to be processed are of manifold length". What is manifold length? Is the term manifold here referred to as a geometric structure like in manifold regularization? Next, the authors wrote "Eq.5", "Eq. 2", and "Eq. 7" when cross-referencing equations, but they should actually be for instance "Eq. (5)" in plain text and "Eq.\~\\\eqref\{label\}" in latex where "\~" inserts a non-breaking space. Please carefully proofread the final version, both English language and latex usage.

**Justification For Why Not Higher Score:**

Writing is not so good. The idea is novel but not so difficult.

**Justification For Why Not Lower Score:**

The studied problem and the obtained results are extremely significant (see Tables 4 and 5).

---

### Decision · Program_Chairs · 2024-01-16

Accept (spotlight)